# Longitudinal biomarker progression and validation for predicting operational tolerance in a prospective multicenter liver transplantation immunosuppression withdrawal trial

Gloria López-Díaz[1◉], María Isabel Sánchez-Lorencio[2◉], Mercedes Iñarrairaegui[3,4], María Luisa González-Diéguez[5], Valle Cadahía[5], Alejandra Otero-Ferreiro[6], María Ángeles Vázquez-Millán[6], Mario Romero-Cristóbal[7,8], Magdalena Salcedo[7,8], Sara Lorente-Pérez[9], Gloria Sánchez-Antolín[10], Jesús de la Peña[11], Pablo Ramírez[12], María P. Pata[13], Alberto Baroja-Mazo[2‡*], José I. Herrero[ID][3,4‡], José Antonio Pons[ID][1,14‡*]

1 Hepatology and Liver Transplant Unit, University Clinical Hospital Virgen de la Arrixaca, Biomedical Research Institute of Murcia (IMIB-Pascual Parrilla), Murcia, Spain, 2 Molecular Inflammation Group, University Clinical Hospital Virgen de la Arrixaca, Biomedical Research Institute of Murcia (IMIB-Pascual Parrilla), Murcia, Spain, 3 Liver Unit, Clínica Universidad de Navarra, Pamplona, Spain, 4 Instituto de Investigación Sanitaria de Navarra (IdiSNA), CIBERehd, Pamplona, Spain, 5 Liver Unit, Hospital Universitario Central de Asturias, Oviedo, Asturias, Spain, 6 Liver Transplant Unit, Complejo Hospitalario A Coruña, A Coruña, Spain, 7 Liver Unit, H.G.U. Gregorio Marañón, Madrid, Spain, 8 CIBERehd, Madrid, Spain, 9 Liver Transplant Unit, Hospital Clínico Universitario Lozano Blesa, Universitiy of Zaragoza and ISS Aragón, Zaragoza, Spain, 10 Hepatology and Liver Transplant Unit, Hospital Universitario Río Hortega, Valladolid, Spain, 11 Pathological Anatomy Unit, University Clinical Hospital Virgen de la Arrixaca, Murcia, Spain, 12 General Surgery Unit, University Clinical Hospital Virgen de la Arrixaca, Biomedical Research Institute of Murcia (IMIB-Pascual Parrilla), Murcia, Spain, 13 Biostatech Advice, Training and Innovation in Biostatistics S.L., Santiago de Compostela, Spain, 14 Internal Medicine Department, Faculty of Medicine, University of Murcia, Campus de Ciencias de la Salud, Murcia, Spain,

◉ These authors contributed equally and share first authorship to this work.
‡ These authors share equal senior last authorship on this work.
* joseapons.imib.arrixaca@gmail.com (JAP); alberto.baroja@ffis.es (AB-M)

## Abstract

Liver transplantation (LT) is a life-saving treatment for end-stage liver disease, but long-term immunosuppression (IS) is associated with significant side effects. Achieving operational tolerance (OT), where the graft is accepted without IS, remains a critical goal. Biomarkers play a pivotal role in understanding the complex mechanisms of OT, enabling personalized treatment strategies and improving patient outcomes. Additionally, machine learning techniques offer powerful tools for identifying predictive biomarkers and optimizing IS withdrawal protocols. This multicenter trial aimed to investigate the longitudinal evolution of genetic biomarkers during IS withdrawal and validate their predictive value for OT in LT recipients. A prospective, multicenter IS withdrawal trial was conducted with 91 LT patients. Tolerant (TOL) and non-tolerant (non-TOL) patients were compared, and longitudinal blood and liver samples were collected to analyze biomarkers. Generalized Additive Mixed Models (GAMMs) and logistic algorithms were employed to assess biomarker associations and predict OT.



**Data availability statement:** All data underlying the findings of this study are fully available within the manuscript and its supplementary materials.

**Funding:** J.A.P. was founded by Instituto de Salud Carlos III (PI17/00489; PI23/00013), co-funded by the European Union. J.I.H. was founded by Instituto de Salud Carlos III (PI17/00699), co-funded by the European Union. A.B-M. was funded by Instituto de Salud Carlos III (PI20/00185; DTS23/00013; PI24/00129), co-funded by the European Union.

**Competing interests:** None of the authors involved in this manuscript have any conflicts of interest to disclose.

Of the 45 patients who completed the trial, 17 (37.8%) achieved OT. Molecular biomarker analysis revealed significant differences between TOL and non-TOL groups. Non-TOL patients exhibited higher baseline methylation of the FOXP3 regulatory T cell-specific demethylated region (TSDR) in whole blood. Longitudinal analysis showed distinct patterns in FOXP3, SENP6, miR31, and miR95 expression between groups. Notably, FOXP3 expression followed a U-shaped trajectory in TOL patients, decreasing during IS withdrawal and increasing post-withdrawal. Machine learning identified several key predictive biomarkers for OT. This study confirms the association between FOXP3 TSDR methylation and OT in LT patients and identifies FEM1C, miR31 and TFRC as promising predictive biomarkers. These findings highlight the potential for personalized IS withdrawal strategies, though further validation in larger cohorts is needed before clinical application.

## Introduction

Liver transplantation (LT) has proven to be a life-saving procedure for patients suffering from end-stage liver disease [1]. The success of this surgical procedure owes much to immunosuppression (IS) that prevents allograft rejection, thus significantly improving graft survival rates and patient outcomes [2]. However, the long-term use of these agents is associated with significant side effects such as increased susceptibility to infections, renal failure, metabolic disorders, and the development of malignancies [3]. As a result, achieving immunological tolerance or operational tolerance (OT), a state in which the immune system or recipients accepts the transplanted organ without the need for continuous IS, has been intensely studied over the past decades [4]. Although the proportion of patients achieving OT is still relatively low [5], these results provide reason for hope and underscore the importance of further research to improve our understanding of tolerance mechanisms and expand the pool of tolerant individuals.

Biomarkers have emerged as invaluable tools in modern medicine, facilitating the diagnosis, prognosis, and monitoring of various diseases [6]. In the field of LT, biomarkers play a crucial role in elucidating the complex mechanisms of OT [7]. These molecular signatures provide insights into the complex interactions between the allograft and the immune system of recipients, allowing clinicians to tailor immunosuppressive regimens and identify patients who may achieve OT [8]. Moreover, different groups have defined several biomarkers trying to distinguish patients who are going to be tolerant from those who are going to reject, including peripheral blood differential expressed genes [9], some intra-hepatic genes involved in iron metabolism [10], or activated Tregs and related microRNAs (miRNAs) [11]. Despite the immense potential of biomarkers, their integration into routine clinical practice has been limited. Conventional biomarkers have not yet demonstrated the desired sensitivity and specificity required for accurate prediction and monitoring of OT [12]. Addressing these challenges urgently requires harnessing the power of advanced technologies and computational approaches. In particular, machine learning algorithms offer promising

opportunities for developing predictive models in LT [13]. By integrating clinical parameters, genetic information, and biomarker profiles, machine learning models have the potential to provide personalized risk assessments, optimize immunosuppressive protocols, and predict long-term outcomes for individual liver transplant recipients [14]. Such models can help identify the key factors associated with successful tolerance induction and improve patient selection for IS weaning.

In the present work, we conducted a prospective trial where 45 liver transplant patients were subjected to a controlled IS withdrawal process, and 17 became tolerant (TOL) and 28 non-tolerant (non-TOL). We studied the evolution of several genetic markers previously related with OT during IS withdrawal process. Furthermore, we found several variables with predictive capacity using machine learning, some of which had previously been validated in a small single-center cohort by our group [15]. These findings aim to improve clinical decision-making in the context of OT in LT.

## Patients and methods

### Study design

In the present study, we analyze the dynamic changes in different genetic parameters based on blood and liver tissue samples collected from LT patients participating in a randomized, controlled, prospective, multicenter and open-label clinical trial involving IS withdrawal conducted across seven hospitals in Spain. The primary long-term aim of this trial was to prospectively assess the impact of IS withdrawal on the clinical evolution of patients, specifically with regard to cardiovascular complications, chronic renal insufficiency, de novo neoplasia, and mortality. The study included adult patients with stable LT and ran from March 2019 to June 2022. The study received approval from the Hospital Clínico Universitario Virgen de la Arrixaca (HCUVA, Murcia, Spain, 2018-7-1-HCUVA) and the national competent authority (AEMPS – Ministry of Health), and registered in EudraCT under the number 2017-004983-37. Prior to enrollment, written consent was obtained from each participant. The study adhered to all ethical principles outlined in the Helsinki Declaration.

Patients, aged 18–75 years and meeting eligibility criteria, were randomly assigned in a 2:1 ratio to either undergo IS drug withdrawal (Study Group) or maintain conventional IS treatment (Control Group), respectively. Continuous monitoring for liver graft dysfunction was conducted. Detailed inclusion criteria, the IS drug weaning protocol, and follow-up procedures are provided in the Supplementary Methods section in S1 File.

### Blood and tissue samples

For the purpose of the present study, six longitudinal blood samples were selected from each LT patient throughout the withdrawal process. The samples included: 1) Basal: collected before the IS withdrawal process, 2) 2M: collected at 2 months after weaning started, 3) 6M: collected after 6 months of weaning, 4) R/TOL: collected at the moment of rejection (R) for the 28 non-TOL patients or 1 month after complete withdrawal for the 17 TOL patients, 5) 6M-postR/-postTOL: collected 6 months after the re-introduction of IS when liver function reached normality for the 28 non-TOL patients (6M-postR) or after complete withdrawal for the 17 TOL patients (6M-postTOL), and 6) 12M-postR/-postTOL: collected one year after rejection or tolerance. On the other hand, three liver biopsies were collected at different times: Basal, R and 12M-postR/-postTOL (Fig 1a).

Different types of samples were obtained from each time point: buffy coat from blood EDTA tubes by centrifugation, and then aliquoted, and frozen at −80°C; Tempus® Blood RNA tubes (ThermoFisher, Waltham, MA, USA) stored at room temperature for 24 hours, then frozen at −20°C until further use; liver tissue fixed in PaxGene Tissue containers (PreAnalitix, Hombrechtikon, Switzerland) and embedded in paraffin.

### Biomarker selection and detection

In this study, we selected a total of 19 variables, which had previously been identified as potential biomarkers for LT tolerance in the literature [9–11], in addition to the expression of seven genes associated with subclinical inflammatory lesions

**a)**

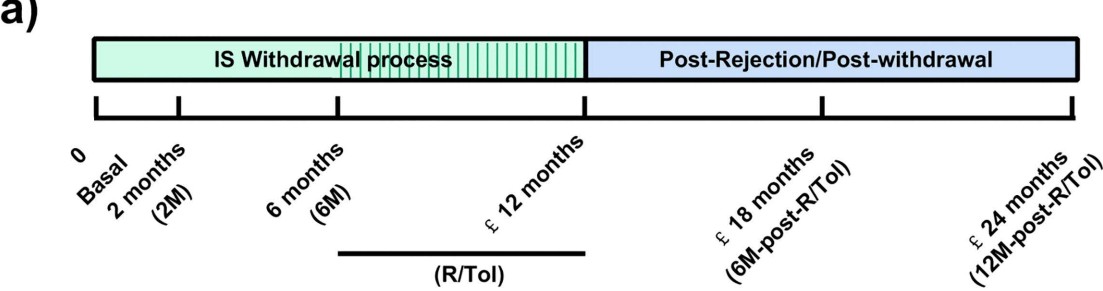

**b)**

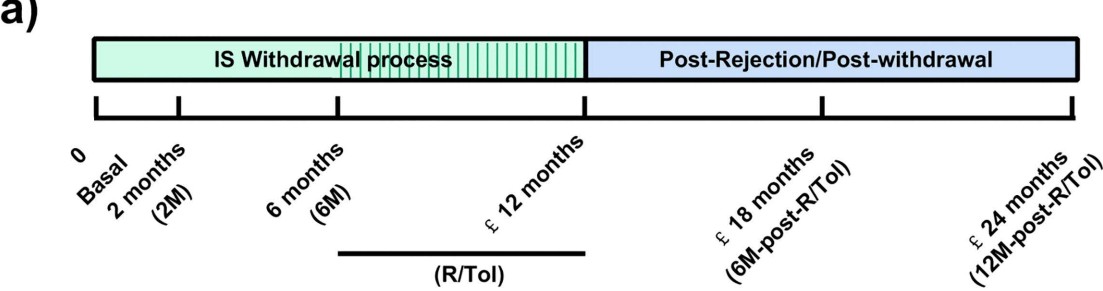

**Fig 1. IS withdrawal protocol. a)** Sample collection schedule. **b)** Clinical trial evolution. Out of the 91 patients enrolled in the study, 52 (57%) underwent a controlled IS withdrawal protocol, and 45 (87%) successfully completed the entire protocol.

in liver tissue [16] (S1 Table). A comprehensive description of these variables and their detection protocols is provided in the Supplementary Methods section in S1 File.

## Statistical analysis

Descriptive statistics were performed for each variable, using the median and mean ± standard deviation (SD) for continuous variables. The Student's t-test or Mann-Whitney test (when variables were not normally distributed) was used to evaluate differences in biomarkers between the TOL and non-TOL groups at different study time points. To assess different features of the longitudinal evolution of the markers, Generalized Additive Mixed Regression Models (GAMMs) were applied [17]. Predictive variables were selected using the regularization and selection method LASSO (Least Absolute Shrinkage and Selection Operator) [18]. The optimal lambda ($\lambda_{min}$) was selected using 10-fold cross-validation. Additionally, to reduce potential overfitting, the largest lambda within one standard error of the minimum error (1-standard-error rule, $\lambda_{1SE}$) was also used. Model performance and potential overfitting were evaluated by analyzing the cross-validation error (mean ± SD) at $\lambda_{min}$ and by constructing learning curves to assess the AUC gap between training and validation. Performance metrics estimated include AUC (95% confidence interval), overall error rate, balanced accuracy, sensitivity, specificity, positive and negative predictive values, and Cohen's kappa. Missing values of predictor variables were imputed using the Predictive Mean Matching method [19].

A comprehensive description of these analysis is available in the Supplementary Methods section in S1 File.

## Results

### Study population

Out of the 91 patients enrolled, 16 were excluded during the screening visit because their biopsy did not meet the Banff Working Group criteria protocol [20]. The remaining 75 patients were randomly divided into two groups. Although not analyzed in the present study, 23 patients comprised the control group (n = 23; G2). Ultimately, 52 patients underwent the controlled IS withdrawal protocol (G1) (Fig 1b). The number of patients per center presented an inclusion disparity, ranging from 2 to 48, with a median of 7 patients by hospital (S2 Table). As expected due to randomization, both groups exhibited similar demographic and clinical characteristics (S3 Table).

### IS withdrawal trial outcome

7 out of the 52 randomized patients in the study group withdrawn the protocol for different reasons (according to the assessment of the clinician, four patients had to be discontinued from the study due to difficulties in traveling to the hospital and concerns about complying with the protocol, one declined to continue in the study, and two died within the trial period) (Fig 1b). Out of the remaining 45 patients, 28 (62.2%) experienced a rejection episode during the withdrawal of IS drugs. These patients were treated by reintroducing the immunosuppressive therapy they had been receiving prior to the weaning protocol, and they were classified as non-tolerant patients (non-TOL). On the other hand, 17 patients (37.8%) successfully completed the drug withdrawal process and achieved tolerance (Fig 1b). TOL patients were defined as those who remained off IS drugs and maintained normal allograft function, without liver lesions for at least one year after IS drug withdrawal.

### Demographic and clinical characteristics of the 45 LT patients who finished the trial

The majority of patients were males (86.7%), and there were no significant differences between the groups (p = 0.809) (Table 1). The median age for both LT patient groups, TOL and non-TOL, was 65 and 61.5 years



**Table 1. Demographic and clinical characteristics of the study population.**

| | Non-Tol (n = 28) | Tol (n = 17) | P |
|---|---|---|---|
| Age [median (range)] | [61.5 (42–73)] | [65 (47–75)] | 0.161a |
| Age at transplantation [median (range)] | [52.5 (22–69)] | [55 (41–69)] | 0.754a |
| Months from transplant to weaning start [median (range)] | [81.5 (37–260)] | [128 (38–233)] | 0.094a |
| Gender (n; %) | | | 0.809b |
| Male | (24; 86) | (15; 88) | |
| Female | (4; 14) | (2; 12) | |
| Co-morbid medical problems (n;%) | | | |
| Diabetes | (15; 54) | (7; 41) | 0.420b |
| Hypertension | (23; 82) | (13; 76) | 0.645b |
| Cardiovascular pathology | (8; 29) | (1; 6) | 0.065b |
| Renal dysfunction | (9; 32) | (6; 35) | 0.718b |
| Diseases (n; %) | | | |
| Alcoholic cirrhosis | (11; 39) | (6; 35) | 0.789b |
| Alcoholic cirrhosis + HCC | (4; 14) | (1; 6) | 0.384b |
| HCV cirrhosis | (6; 21) | (4; 24) | 0.786b |
| HCV cirrhosis + HCC | (2; 7) | (5; 29) | 0.086b |
| Other | (5; 17) | (1; 6) | 0.252b |
| Basal LFTs [median (range)] | | | |
| ALT (U/L) | [17 (5–46)] | [19 (12–25)] | 0.532a |
| ALP (U/L) | [66 (45–125)] | [68 (35–98)] | 0.757a |
| Total Bilirrubin (mg/dL) | [0.57 (0.19–1.89)] | [0.5 (0.2–2.66)] | 0.836a |
| INR | [1.01 (0.95–1.3)] | [1.01 (0.9–1.2)] | 0.592a |
| Main IS-drug (n; %) | | | |
| Cyclosporine A | | (1; 6) | 0.194b |
| Tacrolimus | (9; 32) | (6; 35) | 0.848b |
| Tacrolimus + MMF | (18; 64) | (8; 47) | 0.257b |
| MMF | (1; 4) | (3; 18) | 0.108b |
| Basal tacrolimus concentration (ng/ml) [median (range)] | | | |
| Tacrolimus | [4.5 (2–6.5)] | [2.4 (1–4.5)] | **0.033a** |
| Tacrolimus + MMF | [3.4 (0.6–6)] | [2.4 (1.3–6.8)] | 0.453a |

aT-test. bChi-square test. HCC: Hepatocellular carcinoma; HCV: Hepatitis C virus; LFTs: liver function tests; ALT: Alanine aminotransferase; ALP: Alkaline phosphatase; INR: International normalized ratio; MMF: Mycophenolate mofetil.

old, respectively, with a median age at transplantation of 55 and 52.5 years old, respectively. The TOL group had a longer time from transplant to IS weaning compared to the non-TOL group, although did not reach statistical significance (121.1 ± 58.3 and 92.5 ± 51.7 months, respectively; *p* = 0.094). Most of the transplant patients received tacrolimus as basal IS (89%), typically in dual therapy with mycophenolate mofetil (58%). Nevertheless, significantly more patients in the non-TOL group were treated with tacrolimus than in the TOL group, where 18% received mycophenolate mofetil in monotherapy as baseline IS. A significant difference was found in patients on tacrolimus monotherapy, with the TOL group presenting a lower baseline tacrolimus trough levels compared to the non-TOL patients (Table 1). Likewise, baseline concentration of tacrolimus was significantly different among centers (see S2 Table). However, no correlation was found between the baseline IS concentration and the percentage of tolerance achieved in each center (P = 0.099). The most frequent underlying disease prior to

transplantation was alcoholic cirrhosis (49%), followed by hepatitis C virus infection (38%). The principal co-morbid medical problems detected in these patients were hypertension (80%), diabetes (49%) or renal dysfunction (33%). None of these clinical characteristics, along with liver function tests measured at baseline, showed differences between the groups (Table 1).

## Biomarkers present differences between TOL and non-TOL groups

In our study, several molecular biomarkers that have been previously associated with OT in LT [9–11] were analyzed for each longitudinal sample. In an exploratory data analysis comparing the TOL and non-TOL groups, we observed a significant difference in the forkhead box P3 (*FOXP3*) regulatory T cell-specific demethylated region (TSDR)-methylation ratio (MR) in whole blood at the basal time point. Non-TOL patients showed a higher proportion of methylation compared to TOL patients (Fig 2a). Additionally, the SUMO-specific peptidase 6 (*SENP6*) gene was found to be significantly more expressed in the non-TOL group after two months of weaning, while *FOXP3* expression was higher in the TOL group at this same time-point (Fig 2b). Furthermore, at 12 months after total IS weaning or rejection episode, both *miR31* and *miR95* were found to be significantly more expressed in the non-TOL group (Fig 2c). However, no other significant differences were observed for the rest of the markers at any time-point (S4 Table; S1 Fig in S2 File) or in tissue samples (S5 Table).

## TOL and non-TOL patients present different longitudinal genetic patterns

Basal IS was decreasing in parallel and in a gradual form for both groups of patients in the first 6 months (43.87 ± 23.50% and 53.53 ± 21.24% IS reduction for non-TOL and TOL, respectively; *P* = 0.226) (S2a Fig in S2 File). Rejection was detected after a mean of 364 ± 155 days, with only three patients experiencing rejection before the 6-month time point (at 98, 145 and 167 days, respectively). Non-TOL patients showed a mean IS reduction of 67.91 ± 24.6% at the time of rejection (S2a Fig in S2 File), while alanine aminotransferase (ALT) levels increased up to 120.5 ± 140 U/L (range 54–679) (S2b Fig in S2 File). At this moment, IS was increased, reaching tacrolimus concentrations 12 months post-rejection that were similar to baseline levels (4.54 ± 2.1 ng/mL; *P* = 0.574) (S2a Fig in S2 File), with normal values of ALT (21.5 ± 7.58 U/L; *P* = 0.368) (S2b Fig in S2 File). For TOL group, complete withdrawal moment was reached after 341 ± 69 days (S2a Fig in S2 File), whereas ALT remained under normal values (21.53 ± 6.62; **P = 0.006 compared with non-TOL) (S2b Fig in S2 File).

Likewise, when analyzed longitudinally, both groups presented specific characteristics related to the evolution of different variables over time. In liver tissue samples, *FOXP3* TSDR-MR appeared to be higher 12 months after achieving tolerance, whereas hepcidine antimicrobial peptide *(HAMP)* and guanylate binding protein 2 (*GBP2*) showed the opposite trend (Fig 3a; S6 Table). Nevertheless, *FOXP3* TSDR-MR followed the same trend in non-TOL liver samples, although in that case, only glycoprotein Nmb (*GPNMB*) and matrix metallopeptidase 9 (*MMP9*) showed significantly less expression during the rejection episode and 12 months after (Fig 3b; S6 Table).

Regarding whole blood, we observed significant variations in the expression of *SENP6*, *FOXP3*, and IKAROS family zinc finger 2 (*IKF2*) among different time points in the TOL group (Fig 4a; S6 Table). Conversely, only *IKF2* repeated variations for non-TOL patients, with group-specific differences in the expression of fem-1 homolog C (*FEM1C*), *miR31*, and *FOXP3* TSDR-MR (Fig 4b; S6 Table).

Furthermore, when we estimated the model with time by group interaction for tissue samples, we found that *FOXP3* TSDR-MR significantly increased over time in both TOL and non-TOL patients, with higher ratios in the TOL group, although without significant differences (Fig 5a; S7 Table). For the expression of *HAMP*, the temporal evolution was significant in the TOL group, with the concentration decreasing throughout the study period. For the non-TOL group, the concentration remained constant, but the difference between the curves was not significant. Similarly, for *GBP2* gene

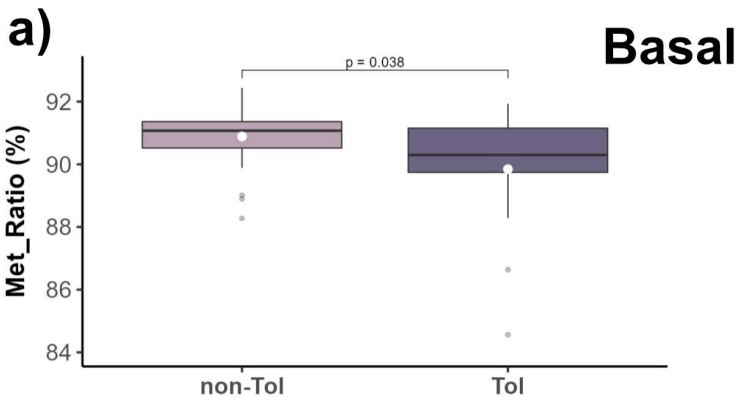

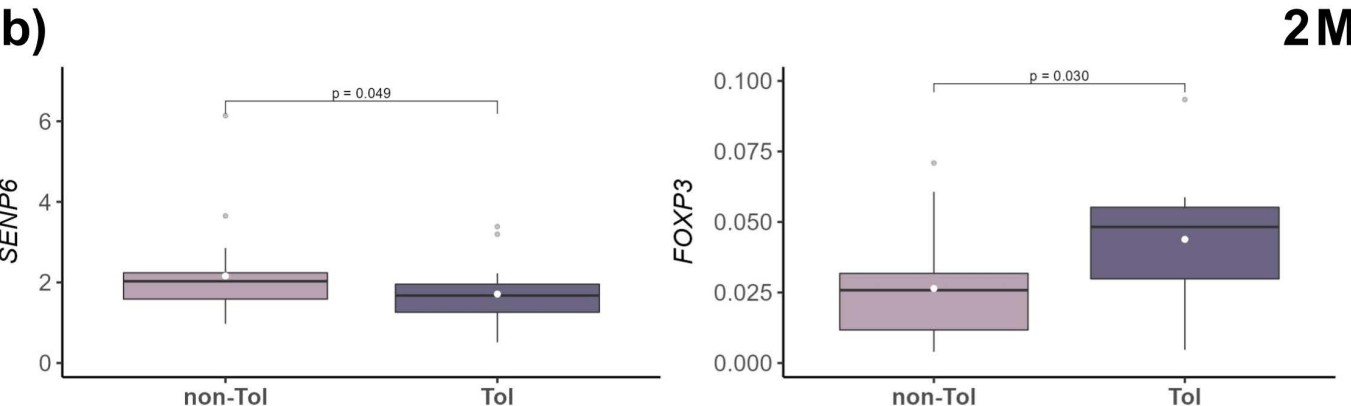

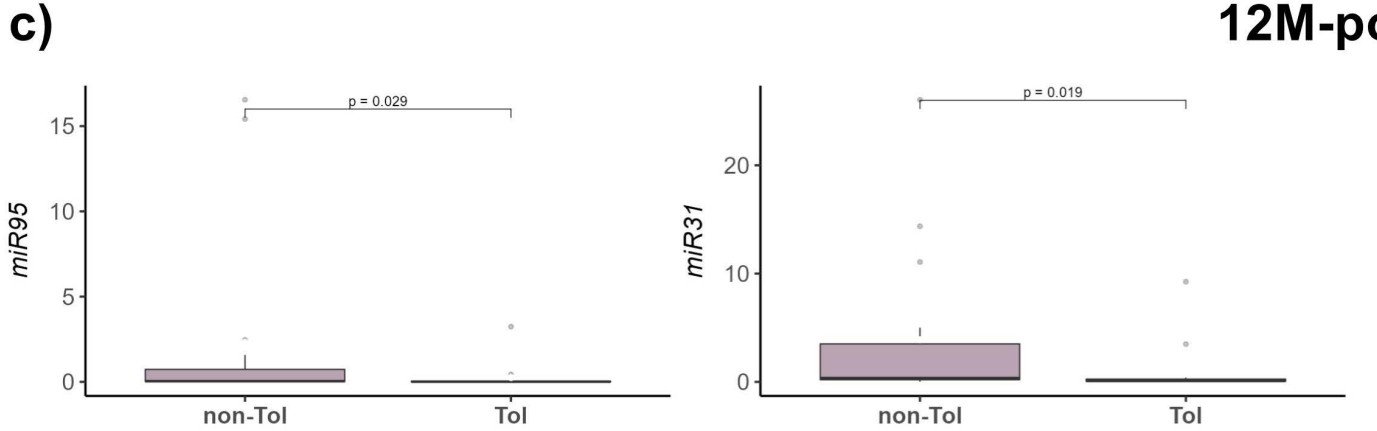

**Fig 2. Key molecular differences between the TOL and non-TOL groups.** Box-plots show *FOXP3*-TSDR methylation at Basal **(a)**, *SENP6* and *FOXP3* expression at 2M **(b)**, and *miR31*/*miR95* expression at 12M (c) in whole blood samples. Median (black line), mean (white dot), and interquartile range are indicated; outliers are shown as dots. Group differences were assessed using the Mann-Whitney-Wilcoxon test.





**Fig 3. Longitudinal tissue variable changes differ between TOL and non-TOL groups.** Plots show variable trajectories across time points in tissue samples from TOL (a) and non-TOL (b) patients. Dots represent individual patients. Lower y-axis section displays estimated means; upper section shows p-values for differences across time points during IS weaning.

expression, the time evolution was significant in the non-TOL group, with the concentration decreasing over the study period, similar to what was found with *GPNMB* expression (Fig 5a; S7 Table).

When the same models were applied to whole blood samples, we found that *FOXP3* expression showed a significant temporal evolution in the TOL group, whereas it remained constant in the rejection group. The concentration decreased



a)

b)

**Fig 4. Longitudinal changes in whole blood differ between TOL and non-TOL groups.** Plots display variable dynamics across time points in whole blood from TOL (a) and non-TOL (b) patients. Dots represent individual samples. Lower y-axis section displays estimated means; upper section shows p-values for differences across time points during IS weaning.

until it reached a minimum around the TOL sample and increased thereafter, showing a characteristic U-shaped pattern (Fig 5b; S7 Table). This situation is in direct contrast to the behavior of *FOXP3* TSDR-MR (Fig 5c; S7 Table). Moreover, the concentration at the Basal sample was significantly lower in the rejection group. The concentration of the *IKF2* marker showed a significant temporal evolution in the TOL group, and although it showed a slight decrease over time in the non-TOL group, it was not significant. In this case, the concentration in the non-TOL group was lower than that in the TOL group between the basal and 2M time points (Fig 5d; S7 Table).

When we applied the linear mixed model with flexible effects to study our groups longitudinally, only *FOXP3* TSDR-MR at the Basal point analyzed in whole blood showed a significant effect on the probability of being in the non-TOL group; a higher percentage of methylation at baseline increased the likelihood of being in the non-TOL group. This pattern, although not significant, was also observed at 2M, 6M, and 12M-post (S3a Fig in S2 File; S8 Table). A similar, but opposite trend was observed for *FOXP3* expression (S3b Fig in S2 File; S8 Table).

### Identification of potential predictive variables for operational tolerance

Starting with a model comprised of 19 biomarkers measured at baseline, along with the demographic and clinical variables such as age, sex, time from transplant to weaning, and tacrolimus concentration, the LASSO method, using $\lambda_{min}$, finally selected 11 predictors with coefficients different from 0 (Fig 6), indicating that these markers have the greatest predictive capacity. These biomarkers included, in order of relevance, *FOXP3* TSDR-MR (analyzed in whole blood – B), *FOXP3* expression (B), transferrin receptor (*TFRC*) expression (analyzed in tissue – T), *FEM1C* expression (B), *miR31* expression (B), *GPNMB* expression (T), suppressor of cytokine signaling 1 (*SOCS1*) expression (T), *HAMP* expression (T), major histocompatibility complex, Class II, DM alpha (*HLA-DMA*) expression (T), and *GBP2* expression (T), in addition to the basal concentration of tacrolimus.

This model achieves strong classification performance (Table 2), with a cross-validated AUC of 0.87 (95% CI: 0.76–0.98), and a mean cross-validation error of 1.09 (SD = 0.02), indicating low variability. When applying the $\lambda_{1SE}$ rule, a more parsimonious model with 4 predictors (*FOXP3 TSDR-MR (B), FOXP3 expression (B)*, *TFRC (T), IKF2 (B)*) resulted in decreased but acceptable performance (Table 2). The training-validation AUC gap of approximately 0.08 in the original model suggests some overfitting, which the $\lambda_{1SE}$ model mitigates through dimensionality reduction.

### Discussion

OT induction in LT is a challenging but feasible process that could promote significant patient benefits by avoiding serious side effects. Although still ongoing, we initiated a multicenter, randomized, prospective, and controlled clinical trial of IS weaning to assess its long-term efficacy. As in recent studies [21], 18% of patients presented altered histopathology at baseline, despite normal biochemistry, leading to their exclusion from IS withdrawal [20]. Nevertheless, approximately 40% of patients undergoing IS withdrawal achieved OT, surpassing the average success rate reported elsewhere [5], suggesting that careful candidate selection may enhance success [21,22]. In line with previous findings from our group and others [23,24], the interval between transplantation and IS withdrawal tended to be longer in TOL patients, although it did not reach statistical significance (*p* = 0.094). These observations supports the trial's design and the robustness of the data. Additionally, TOL patients receiving tacrolimus monotherapy had a significantly lower concentration of tacrolimus at inclusion. Lower concentrations may promote great immune activation, which could support tolerance induction [25]. Accordingly, in a pediatric trial where all liver transplant recipients had very low basal tacrolimus trough levels, 100% achieved OT [26]. Notably, none of the TOL patients required reintroduction of IS during extended follow-up. Only two

a)



b)

c)

d)

**Fig 5. Group-specific trajectories of markers over time.** Generalized Additive Mixed Models (GAMMs) show marker evolution by group. Dotted lines indicate mean time points per interval; corresponding mean tacrolimus levels are shown in whole blood panels (b–d, left plots). Blue: TOL; red: non-TOL. Shaded blue areas mark intervals with significant group differences (b-d, right plots). Horizontal line at 0 indicates no difference; values above/below reflect higher/lower levels in non-TOL vs. TOL. Shaded bands show 95% CI.

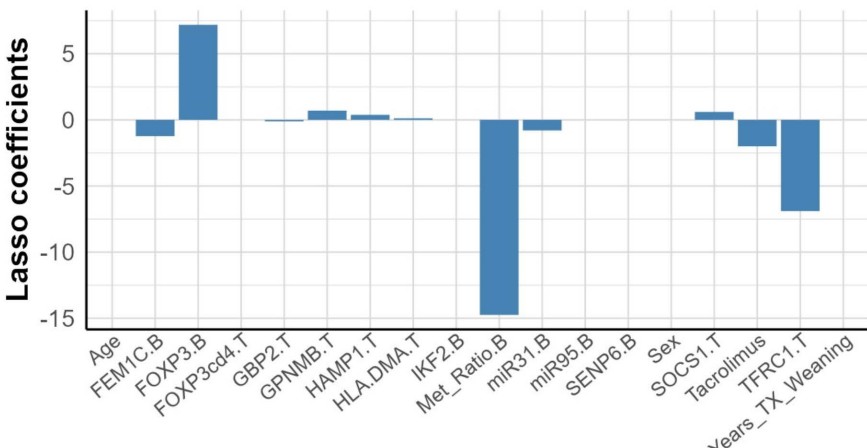

**Fig 6. Selected variables to detect potential tolerant patients.** A bar plot illustrates the coefficients estimated by the LASSO method (Y-axis) for each of the candidate predictors (X-axis).

**Table 2. Classification performance metrics of the LASSO logistic regression models selected using the minimum cross-validated error criterion ($\lambda_{min}$) and the one-standard-error rule ($\lambda_{1SE}$).**

| Model* | Error | Balanced Accuracy | Sensitivity | Specificity | Positive Predictive Value | Negative Predictive Value | Kappa |
|---|---|---|---|---|---|---|---|
| $\lambda_{min}$ (11 predictors) | 13.3 | 0.858 | 0.824 | 0.893 | 0.824 | 0.893 | 0.716 |
| $\lambda_{1SE}$ (4 predictors) | 34.4 | 0.686 | 0.812 | 0.561 | 0.529 | 0.831 | 0.337 |

*The $\lambda_{min}$ model includes 11 predictors, while the more parsimonious $\lambda_{1SE}$ model includes 4 predictors. Metrics include overall error rate, balanced accuracy, sensitivity, specificity, positive and negative predictive values, and Cohen's kappa.

patients experienced mild, transient elevations in liver enzymes, which resolved spontaneously within one month, likely reflecting minor, self-limiting inflammatory responses rather than true rejection episodes [24].

Taking advantage of the opportunity to collect valuable longitudinal samples during the clinical trial, we analyzed the evolution of established biomarkers throughout the process of tolerance. Among the genes and miRNAs previously associated with OT, the most consistently related biomarker was regulatory T cells (Tregs). FOXP3, the master transcription factor of Tregs, controls key genes for their differentiation and function [27]. Complete demethylation of the *FOXP3* TSDR region is restricted to Tregs [28], making it useful for quantifying them in blood or tissue across diseases [29]. We observed that *FOXP3* expression and TSDR-MR followed opposite trends, suggesting that lower methylation corresponds to higher expression [11]. This suggest a higher proportion of a stable, active Tregs, as FOXP3 stability is crucial for their suppressive function [30,31]. However, we cannot determine whether this is due to an increase in Tregs or a decrease in T effector cells.

While previous studies from our group showed no group differences in whole blood FOXP3 TSDR-MR [11,15], in this trial, we found significant differences, possibly due to using pyrosequencing, a more sensitive technique [32], and analyzing eight CpG sites compared to two in previous methyl qRT-PCR-based studies. Interestingly, *FOXP3* TSDR-MR increased after treatment in both groups, a finding of uncertain significance given the lack of previous tissue-level studies

in this context. Furthermore, no correlation between tissue *FOXP3* TSDR-MR and *FOXP3* expression has been observed [33], and not all Treg functionality relies on *FOXP3* TSDR demethylation [34].

Strikingly, *FOXP3* expression in the TOL group followed a U-shaped curve, decreasing until complete IS withdrawal, then increasing, while remaining unchanged in non-TOL patients. This could reflect subclinical liver inflammation in TOL patients after IS withdrawal, as previously described [35–37]. This contrasts with our 2008 findings [38], likely due to methodological differences: frozen peripheral blood mononuclear cells (PBMCs) in 2008 vs. stabilized whole blood RNA here, which better captures immune gene expression [39]. Nevertheless, these findings highlight three important points: a) technical approaches and sample types impact findings; b) Tregs, via *FOXP3* expression or TSDR-MR, remain pivotal to OT in LT [40]; and c) although IS drugs can influence blood gene expression profiles post-LT [41], *FOXP3* expression appears unaffected.

In the era of artificial intelligence, valuable biomarkers should be exploited to develop predictive models using machine learning (ML). This approach holds great potential for improving clinical decision-making [42] and could transform the field of OT in LT. Although predictive models to detect OT should be a priority [43,44], few ML studies have been published, and most rely on retrospective samples [15,45]. Our current work builds on prior findings by associating biomarkers of different types, some previously validated in our small, single-center cohort [15], with OT.

By integrating biomarkers from tissue and whole blood, at both DNA and RNA levels, along with clinical/demographic parameters, we identified candidate predictors at baseline. Beyond *FOXP3* expression and methylation, other relevant contributors included intra-graft *TFRC* and blood *FEM1C* and *miR31 expression*. Both FEM-1 homolog C and *miR31* showed strong predictive capacity in our previous work [15], reinforcing their importance in OT. *FEM1C,* though still poorly understood in this context, belongs to the highly conserved VHL-box proteins family, involved in cell cycle regulation and cell proliferation via the Cullin 2-RING E3 ubiquitin ligase complex [46], and its downregulation has been linked to colorectal cancer progression [47]. *miR31,* known to regulate Tregs, is downregulated in these cells and suppresses *FOXP3* expression by binding to its 3' UTR [48–50]. In our model, both biomarkers were more influential in the non-TOL group, suggesting a role in tolerance failure. Similarly, TFRC, a key regulator of iron metabolism, [51], was upregulated in non-TOL patients and shown to correlate with low serum ferritin levels [10].

Nonetheless, this has limitations. The sample size was constrained, largely due to the negative impact of the COVID-19 pandemic on non-COVID-19 research activities [52]. We enrolled 91 patients across 7 different Spanish hospitals, with 45 patients completing the IS withdrawal protocol. Despite this, the trial remains one of the largest European studies in adult LT tolerance [23]. Obviously, small sample sizes impacts the consistency of statistical analyses and predictive modeling [53]. However, the use of regularization methods, such as the LASSO approach [18], have been shown to be robust under high-dimensional settings with limited sample size, allowing for low-biased parameter estimates and effective feature selection [54]. Nevertheless, given the inherent risk of overfitting when selecting multiple predictors in small samples, we compared the standard minimum cross-validated error criterion ($\lambda_{min}$) with the more conservative one-standard-error rule ($\lambda_{1SE}$). The modest training–validation AUC gap and low cross-validation error variance in the $\lambda_{min}$ model suggest limited overfitting and robust performance estimates. By transparently reporting both models and their trade-offs, we align with best practices for high-dimensional, small-cohort biomedical studies [55,56]. Notably, both *FOXP3* TSDR-MR and *FOXP3* expression remained key predictors in the reduced $\lambda_{1SE}$ model, reinforcing their stable association with operational tolerance. These findings emphasize the importance of assessing model robustness beyond selection metrics and support the predictive value of our main biomarker across regularization approaches.

Notably, some biomarkers previously validated in a smaller single-center cohort [15] again demonstrated predictive capacity in this multi-center setting. Still, these findings require further validation in new cohorts, ideally through large-scale multicenter international trials, before clinical integration. As recently highlighted, re-validating biomarkers for liver OT is particularly challenging [57]. Additionally, some biomarkers were assessed in tissue biopsies, which poses

translational limitations due to procedural invasiveness [58,59]. However, baseline liver biopsy remains essential in IS weaning protocol [20], ethically supporting biomarker analysis in this context.

From these limitations, our study presents key strengths. It is the first prospective, longitudinal, multicenter trial to systematically monitor molecular biomarkers (epigenetic, transcriptomic, and miRNA) in both blood and tissue during IS withdrawal in LT. The multicenter design, harmonized protocols, and dual-sample analysis enhance the robustness and clinical relevance of our results. Moreover, independent multicenter validation of previously reported biomarkers strengthens their predictive utility and supports their potential role in personalizing IS strategies.

From a clinical perspective, our findings mark a step toward precision medicine in LT. The proposed biomarkers may help stratify patients by OT potential, enabling more targeted IS tapering and reducing drug-related toxicity. The analytic methods, including pyrosequencing for methylation and RT-qPCR for miRNAs and gene expression analysis, are widely available in clinical labs and compatible with routine workflows. Costs per sample are moderate, and bulk testing or centralized services could further reduce expenses, making implementation feasible outside research settings. With continued validation and standardization, this biomarker panel could become a practical tool for real-world LT management.

## Supporting information

**S1 File. Supplementary methods.**
(DOCX)

**S2 File. Supplementary Figures.**
(DOCX)

**S1 Table. Biomarkers analyzed.**
(DOCX)

**S2 Table. Patients included per center.**
(DOCX)

**S3 Table. Comparison of demographic and clinical characteristics between control and study groups.**
(DOCX)

**S4 Table. Statistical analysis of different variables evaluated in whole blood between TOL and non-TOL groups at different time points.**
(DOCX)

**S5 Table. Statistical analysis of different variables evaluated in liver tissue between TOL and non-TOL groups at different time points.**
(DOCX)

**S6 Table. Statistical analysis of different variables evaluated longitudinally in liver tissue and whole blood along the withdrawal protocol.**
(DOCX)

**S7 Table. Statistics from GAMMs of different variables evaluated longitudinally in liver tissue and whole blood along the withdrawal protocol.**
(DOCX)

**S8 Table. Statistics from linear mixed model with flexible effects of different variables evaluated longitudinally in liver tissue and whole blood along the withdrawal protocol.**
(DOCX)

## Acknowledgments

This work was supported by the Instituto de Salud Carlos III (ISCIII) Platform for Clinical Research Support – SCReN (Spanish Clinical Research Network). It also received support from the Clinical Trials Platform of the Instituto Murciano de Investigación Biosanitaria (IMIB-Pascual Parrilla). We are particularly grateful for the generous contribution of the patients and the collaboration of Biobank Network of the Region of Murcia, BIOBANC-MUR, registered on the Registro Nacional de Biobancos with registration number B.0000859. Likewise, the authors thank the Genomics and Pathology Core at the IMIB-Pascual Parrilla for support.

## Author contributions

**Conceptualization:** Alberto Baroja-Mazo, José I. Herrero, José Antonio Pons.

**Data curation:** María Isabel Sánchez-Lorencio, Mercedes Iñarrairaegui, María Luisa González-Diéguez, Valle Cadahía, Alejandra Otero-Ferreiro, María Ángeles Vázquez-Millán, Mario Romero-Cristóbal, Magdalena Salcedo, Sara Lorente-Pérez, Gloria Sánchez-Antolín, Jesús de la Peña, Pablo Ramírez, María P. Pata, Alberto Baroja-Mazo, José Antonio Pons.

**Formal analysis:** María Isabel Sánchez-Lorencio, Mercedes Iñarrairaegui, María Luisa González-Diéguez, Valle Cadahía, Alejandra Otero-Ferreiro, María Ángeles Vázquez-Millán, Mario Romero-Cristóbal, Magdalena Salcedo, Sara Lorente-Pérez, Gloria Sánchez-Antolín, Jesús de la Peña, Pablo Ramírez, María P. Pata, Alberto Baroja-Mazo, José I. Herrero.

**Funding acquisition:** Alberto Baroja-Mazo, José I. Herrero, José Antonio Pons.

**Investigation:** Gloria López-Díaz, María Isabel Sánchez-Lorencio, Mercedes Iñarrairaegui, María Luisa González-Diéguez, Valle Cadahía, Alejandra Otero-Ferreiro, María Ángeles Vázquez-Millán, Mario Romero-Cristóbal, Magdalena Salcedo, Sara Lorente-Pérez, Gloria Sánchez-Antolín, Jesús de la Peña, Pablo Ramírez, Alberto Baroja-Mazo, José I. Herrero, José Antonio Pons.

**Methodology:** María P. Pata, Alberto Baroja-Mazo, José I. Herrero, José Antonio Pons.

**Project administration:** Alberto Baroja-Mazo, José I. Herrero, José Antonio Pons.

**Supervision:** Alberto Baroja-Mazo, José I. Herrero, José Antonio Pons.

**Validation:** Gloria López-Díaz, María P. Pata, Alberto Baroja-Mazo, José I. Herrero, José Antonio Pons.

**Writing – original draft:** Alberto Baroja-Mazo, José I. Herrero, José Antonio Pons.

**Writing – review & editing:** Gloria López-Díaz, Alberto Baroja-Mazo, José I. Herrero, José Antonio Pons.

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
