## [Decision Letter · Decision Letter 0]

30 Apr 2025

Dear Dr. Pons,

Thank you for submitting your manuscript to PLOS ONE. After careful consideration, we feel that it has merit but does not fully meet PLOS ONE’s publication criteria as it currently stands. Therefore, we invite you to submit a revised version of the manuscript that addresses the points raised during the review process.

We look forward to receiving your revised manuscript.

Kind regards,

Mohammad Reza Fattahi, M.D., M.P.H.

Academic Editor

PLOS ONE

**Journal Requirements:**

1. When submitting your revision, we need you to address these additional requirements. Please ensure that your manuscript meets PLOS ONE's style requirements, including those for file naming. The PLOS ONE style templates can be found at https://journals.plos.org/plosone/s/file?id=wjVg/PLOSOne_formatting_sample_main_body.pdf and https://journals.plos.org/plosone/s/file?id=ba62/PLOSOne_formatting_sample_title_authors_affiliations.pdf 2. Thank you for stating the following financial disclosure: J.A.P. was founded by Instituto de Salud Carlos III (PI17/00489; PI23/00013), co-funded by the European Union. J.I.H. was founded by Instituto de Salud Carlos III (PI17/00699), co-funded by the European Union. A.B-M. was funded by Instituto de Salud Carlos III (PI20/00185; DTS23/00013; PI24/00129), co-funded by the European Union.  Please state what role the funders took in the study.  If the funders had no role, please state: "The funders had no role in study design, data collection and analysis, decision to publish, or preparation of the manuscript." If this statement is not correct you must amend it as needed. Please include this amended Role of Funder statement in your cover letter; we will change the online submission form on your behalf. 3. Thank you for stating the following in the Acknowledgments Section of your manuscript: This work was supported by the Instituto de Salud Carlos III (ISCIII) Platform for Clinical Research Support - SCReN (Spanish Clinical Research Network), funded by ISCIII through project PI17/00489 and co-funded by the European Regional Development Fund (ERDF) “A way of making Europe.” It also received support from the Clinical Trials Platform of the Instituto Murciano de Investigación Biosanitaria (IMIB-Pascual Parrilla). We are particularly grateful for the generous contribution of the patients and the collaboration of Biobank Network of the Region of Murcia, BIOBANC-MUR, registered on the Registro Nacional de Biobancos with registration number B.0000859. BIOBANC-MUR and supported by the ISC III (PT20/00109), by “IMIB-Pascual Parrilla” and by “Consejeria de Salud de la Comunidad Autónoma de la Región de Murcia”. Likewise, the authors thank the Genomics and Pathology Core at the IMIB-Pascual Parrilla for support. We note that you have provided funding information that is not currently declared in your Funding Statement. However, funding information should not appear in the Acknowledgments section or other areas of your manuscript. We will only publish funding information present in the Funding Statement section of the online submission form. Please remove any funding-related text from the manuscript and let us know how you would like to update your Funding Statement. Currently, your Funding Statement reads as follows: J.A.P. was founded by Instituto de Salud Carlos III (PI17/00489; PI23/00013), co-funded by the European Union. J.I.H. was founded by Instituto de Salud Carlos III (PI17/00699), co-funded by the European Union. A.B-M. was funded by Instituto de Salud Carlos III (PI20/00185; DTS23/00013; PI24/00129), co-funded by the European Union.  Please include your amended statements within your cover letter; we will change the online submission form on your behalf. 4. In the online submission form, you indicated that “The data that support the findings of this study are available from the corresponding author upon reasonable request. Some data may not be made available because of privacy or ethical restrictions.” All PLOS journals now require all data underlying the findings described in their manuscript to be freely available to other researchers, either a. In a public repository, b. Within the manuscript itself, or c. Uploaded as supplementary information.This policy applies to all data except where public deposition would breach compliance with the protocol approved by your research ethics board. If your data cannot be made publicly available for ethical or legal reasons (e.g., public availability would compromise patient privacy), please explain your reasons on resubmission and your exemption request will be escalated for approval.

**Additional Editor Comments:**

Please shrink your manuscript, add the clinical applicability and double check your references.

Reviewers' comments:

Reviewer's Responses to Questions

**Comments to the Author**

1. Is the manuscript technically sound, and do the data support the conclusions?

Reviewer #1: Yes

Reviewer #2: Yes

2. Has the statistical analysis been performed appropriately and rigorously?

Reviewer #1: I Don't Know

Reviewer #2: Yes

3. Have the authors made all data underlying the findings in their manuscript fully available?

Reviewer #1: No

Reviewer #2: Yes

4. Is the manuscript presented in an intelligible fashion and written in standard English?

Reviewer #1: Yes

Reviewer #2: Yes

**Reviewer #1:**  Dear authors

Your manuscript is very interesting about a clinical crucial topic in liver transplantation

I have very few concerns I included only comments to improve your manuscript

1. I would like to know the costs of analysis and if could be done in a nom-experiemntal setting to know if your nice study could be available to any LT Team

2. I think the figures text are too long please try to short it

3. Discussion is nice but I think that is also too long please try to make it short

4. You mentioned limitations but include also strenghts.

**Reviewer #2:**  This study is very interesting and sheds light on future research lines to predict OT in liver transplantation. There are some points to highlight to consider this study for publication:

1. From the 17 patients who achieved tolerance with normal liver function over one year, it would be interesting to know how many keep tolerant on the long run and how many had a rejection episode since the first year. Consider adding this clinical information.

2. In the tolerance group, have you seen biomarker differences in patients with Mycophenolate mofetil on monotherapy from patients with tacrolimus based regimen. Consider adding this information if has been analyzed.

3. As mentioned in limitations, which regularization methods have been used to counteract small sample size? Please mention them If applicable.

4. Some references are incomplete (23, 27) (missing page numbers).

**Do you want your identity to be public for this peer review?** For information about this choice, including consent withdrawal, please see our Privacy Policy

Reviewer #1: No

Reviewer #2: No

---

## [Author Response · Author response to Decision Letter 1]

28 May 2025

Manuscript Reference PONE-D-25-05354

Point-by-point Rebuttal Letter

We especially thank the reviewers and editors for their comments. We have revised the manuscript in accordance with their suggestions and we believe these revisions will satisfy the issues raised by the referees. Changes in the revised manuscript are displayed as "tracked changes". Also, a clean version of the manuscript has been submitted.

Journal Requirements:

We have revised the manuscript in accordance with all PLOS ONE style requirements.

J.A.P. was founded by Instituto de Salud Carlos III (PI17/00489; PI23/00013), co-funded by the European Union. J.I.H. was founded by Instituto de Salud Carlos III (PI17/00699), co-funded by the European Union. A.B-M. was funded by Instituto de Salud Carlos III (PI20/00185; DTS23/00013; PI24/00129), co-funded by the European Union.

We have incorporated the updated Role of Funder statement in the cover letter, as requested.

This work was supported by the Instituto de Salud Carlos III (ISCIII) Platform for Clinical Research Support - SCReN (Spanish Clinical Research Network), funded by ISCIII through project PI17/00489 and co-funded by the European Regional Development Fund (ERDF) “A way of making Europe.” It also received support from the Clinical Trials Platform of the Instituto Murciano de Investigación Biosanitaria (IMIB-Pascual Parrilla). We are particularly grateful for the generous contribution of the patients and the collaboration of Biobank Network of the Region of Murcia, BIOBANC-MUR, registered on the Registro Nacional de Biobancos with registration number B.0000859. BIOBANC-MUR and supported by the ISC III (PT20/00109), by “IMIB-Pascual Parrilla” and by “Consejeria de Salud de la Comunidad Autónoma de la Región de Murcia”. Likewise, the authors thank the Genomics and Pathology Core at the IMIB-Pascual Parrilla for support.

J.A.P. was founded by Instituto de Salud Carlos III (PI17/00489; PI23/00013), co-funded by the European Union. J.I.H. was founded by Instituto de Salud Carlos III (PI17/00699), co-funded by the European Union. A.B-M. was funded by Instituto de Salud Carlos III (PI20/00185; DTS23/00013; PI24/00129), co-funded by the European Union.

In the revised manuscript, we have ensured that the Acknowledgements section no longer contains any funding information.

4. In the online submission form, you indicated that “The data that support the findings of this study are available from the corresponding author upon reasonable request. Some data may not be made available because of privacy or ethical restrictions.”

We apologize for the confusion. Please note that all relevant data have been included in the main manuscript and/or the supplementary files.

Additional Editor Comments:

Please shrink your manuscript, add the clinical applicability and double check your references.

We sincerely thank the Editor for the additional comments and constructive guidance. In response, we have carefully revised the manuscript as follows:

Manuscript length: We have carefully reduced the length of the text, focusing on improving clarity and eliminating redundancy, particularly in the Discussion and Figure Legends, while preserving the scientific integrity and core findings.

Clinical applicability: We have expanded the discussion and conclusions to better articulate the clinical relevance of our findings. In particular, we included a dedicated paragraph outlining the potential for real-world implementation, cost considerations, and accessibility of the biomarker assays. We also responded to reviewer concerns about long-term follow-up and post-weaning clinical outcomes to contextualize the results.

References: All citations have been thoroughly reviewed and revised for completeness and accuracy. Where applicable, we have included DOI and PMID identifiers to align with PLOS ONE’s formatting requirements.

We have also addressed all reviewer comments point-by-point and incorporated the requested revisions into the main manuscript. We hope these changes meet the journal’s standards and expectations.

Thank you for considering our revised submission.

Reviewer #1:

Dear authors,

Your manuscript is very interesting about a clinical crucial topic in liver transplantation

I have very few concerns I included only comments to improve your manuscript

I would like to know the costs of analysis and if could be done in a no-experimental setting to know if your nice study could be available to any LT Team

We appreciate the reviewer’s interest in the real-world applicability of our findings. As suggested, we have expanded the Discussion to address this point explicitly. We now clarify that the molecular techniques used (pyrosequencing and RT-qPCR) are compatible with most hospital or reference laboratories and rely on commercially available kits. The per-sample cost is moderate and scalable, making implementation feasible in non-experimental clinical settings. These additions can be found in the final paragraph of the Discussion section, starting with “From a clinical perspective...”.

I think the figures text are too long please try to short it

We thank the reviewer for this suggestion. In response, we have carefully revised and shortened the figure legends throughout the manuscript, particularly Figures 2–5, to enhance clarity and conciseness while preserving essential information. We believe the revised legends now better align with the journal’s standards and improve overall readability.

Discussion is nice but I think that is also too long please try to make it short

We thank the reviewer for this constructive comment. In response, we have carefully revised the Discussion section and reduced its length by approximately 15–20%, while ensuring that the scientific content, clarity, and references were preserved. We streamlined the text by condensing repetitive or overly detailed sections, merging related ideas, and simplifying complex phrasing where possible. We hope the revised version better meets the journal’s expectations for conciseness and enhances overall readability.

You mentioned limitations but include also strenghts.

We thank the reviewer for this valuable suggestion. In response, we have now included a dedicated paragraph at the end of the Discussion section that highlights the strengths of our study. In particular, we emphasize its prospective multicenter design, the longitudinal and dual-compartment (blood and tissue) biomarker analysis, and the validation of previously published biomarkers in an independent multicenter cohort. These features support the robustness and translational value of our findings. Please see the revised manuscript, Discussion section, paragraph starting with “Despite these limitations...”.

Reviewer #2:

This study is very interesting and sheds light on future research lines to predict OT in liver transplantation. There are some points to highlight to consider this study for publication:

From the 17 patients who achieved tolerance with normal liver function over one year, it would be interesting to know how many keep tolerant on the long run and how many had a rejection episode since the first year. Consider adding this clinical information.

We thank the reviewer for this important question. None of the 17 patients who achieved operational tolerance at one year have required reintroduction of immunosuppression during the follow-up period. Two patients experienced isolated, mild elevations in liver enzymes (GPT levels of 70 and 50 IU/L, respectively) at 18 and 24 months post-IS withdrawal. These episodes resolved spontaneously within one month without treatment and were not accompanied by clinical symptoms or histological confirmation of rejection. These transient abnormalities are not uncommon in patients off IS and may reflect minor, self-limited inflammatory episodes rather than true rejection. Given the limited follow-up time and the absence of significant clinical events, all 17 patients remain off IS and continue to be monitored regularly. We have added this information to the discussion section of the Manuscript (see page 19, lines 373-376: “Notably, none of the TOL patients required reintroduction of IS during extended follow-up. Only two patients experienced mild, transient elevations in liver enzymes, which resolved spontaneously within one month, likely reflecting minor, self-limiting inflammatory responses rather than true rejection episodes [24]”).

In the tolerance group, have you seen biomarker differences in patients with Mycophenolate mofetil on monotherapy from patients with tacrolimus based regimen. Consider adding this information if has been analyzed.

We appreciate the reviewer’s insightful suggestion. In our cohort, only three patients within the tolerance group were receiving monotherapy with MMF, while the vast majority were on a tacrolimus-based regimen. Given the very limited number of MMF monotherapy cases, we believe that a comparative statistical analysis would lack sufficient power and could lead to misleading interpretations. For this reason, we have opted not to include subgroup comparisons based on baseline immunosuppressive regimen. However, we agree that future studies with larger sample sizes could address this interesting question more robustly.

As mentioned in limitations, which regularization methods have been used to counteract small sample size? Please mention them If applicable.

We thank the reviewer for pointing out the need to clarify this methodological aspect. As noted in the Methods section and detailed in the Supplementary Methods, we applied LASSO (Least Absolute Shrinkage and Selection Operator) regularization to reduce model overfitting and handle high-dimensional data in the context of our limited sample size. LASSO was used during the predictive model development phase to perform variable selection and penalize less informative features, thus ensuring stable and interpretable models. We have now explicitly mentioned this in the Manuscript to improve clarity (Discussion, page 22, lines 440-443: “However, the use of regularization methods, such as the LASSO approach [18], have been shown to be robust under high-dimensional settings with limited sample size, allowing for low-biased parameter estimates and effective feature selection [54]”).

Some references are incomplete (23, 27) (missing page numbers).

We appreciate the reviewer’s careful assessment of the reference list. Following PLOS ONE's guidelines and to ensure consistency and completeness, we have revised all references in the manuscript. Specifically, we have included both the DOI for all articles and the PMID where applicable, in accordance with the journal’s format requirements. This approach ensures persistent and unambiguous access to all cited sources, even when page numbers are not available. We have double-checked the full reference list to ensure compliance with these standards.

---

## [Decision Letter · Decision Letter 1]

29 Jun 2025

Dear Dr. Pons,

Thank you for submitting your manuscript to PLOS ONE. After careful consideration, we feel that it has merit but does not fully meet PLOS ONE’s publication criteria as it currently stands. Therefore, we invite you to submit a revised version of the manuscript that addresses the points raised during the review process.

We look forward to receiving your revised manuscript.

Kind regards,

Mohammad Reza Fattahi, M.D., M.P.H.

Academic Editor

PLOS ONE

Journal Requirements:

Reviewers' comments:

Reviewer's Responses to Questions

**Comments to the Author**

Reviewer #1: All comments have been addressed

Reviewer #2: All comments have been addressed

Reviewer #3: (No Response)

2. Is the manuscript technically sound, and do the data support the conclusions?

Reviewer #1: Yes

Reviewer #2: Yes

Reviewer #3: Partly

3. Has the statistical analysis been performed appropriately and rigorously?

Reviewer #1: I Don't Know

Reviewer #2: Yes

Reviewer #3: No

4. Have the authors made all data underlying the findings in their manuscript fully available?

Reviewer #1: No

Reviewer #2: Yes

Reviewer #3: No

5. Is the manuscript presented in an intelligible fashion and written in standard English?

Reviewer #1: Yes

Reviewer #2: Yes

Reviewer #3: Yes

Reviewer #1: DEAR AUTHORS:

You have answered my questions adequately.

The manuscript has improved with the changes you performed

Reviewer #2: I have no more comments to add. The last version is clear. Congratulations to the authors for the research.

Reviewer #3: My comments are listed below.

Line 213-215, TOL group had a longer time from transplant to IS weaning compared to non-

TOL group, although it did not reach statistical significance (121.1 ± 58.3 and 92.5± 51.7 months, respectively; p = 0.094), such obvious difference in length of using IS may have influence on the outcome. Such a factor may be included in the list of candidate predictors for the predictive models.

A total of 11 predictors were selected by LASSO, how reliable would these selected predictors be with the very limited sample size of 45 (17 TOL vs. 28 non-TOL)? The model would be overfitted by including 11 predictors in a model with n=45. To avoid overfitting and remove less impactful predictors, you may use a larger lambda as minimum lambda + 1SE. The final model would have less concern of overfitting with 3-4 selected predictors based on n=45.

The predictive capacity and performance of the model with selected predictors should be examined and reported.

**Do you want your identity to be public for this peer review?** For information about this choice, including consent withdrawal, please see our Privacy Policy

Reviewer #1: No

Reviewer #2: No

Reviewer #3: No

---

## [Author Response · Author response to Decision Letter 2]

2 Aug 2025

Dear Editor and Reviewers,

We sincerely thank Reviewers 1 and 2 for their constructive feedback and for accepting our responses and revisions to the previous comments raised. We are now addressing the questions and concerns expressed by the new Reviewer 3, as indicated by the Editor. Below, we provide a point-by-point response to each of the reviewer’s remarks, incorporating clarifications and modifications to the manuscript where applicable. Changes in the revised manuscript are displayed as "tracked changes". Also, a clean version of the manuscript has been submitted.

Journal Requirements:

We have carefully reviewed the entire reference list to ensure its accuracy and completeness. All references are up-to-date and relevant to the study’s context.

Reviewer #3:

My comments are listed below.

Line 213-215, TOL group had a longer time from transplant to IS weaning compared to non-TOL group, although it did not reach statistical significance (121.1 ± 58.3 and 92.5± 51.7 months, respectively; p = 0.094), such obvious difference in length of using IS may have influence on the outcome. Such a factor may be included in the list of candidate predictors for the predictive models.

We sincerely appreciate the reviewer’s attention to this detail. We would like to clarify that "time from transplant to weaning" was indeed included as a candidate predictor in the LASSO model from the outset, as stated in the "Identification of potential predictive variables for operational tolerance" section (first paragraph):

"Starting with a model comprised of 19 biomarkers measured at baseline, along with the demographic and clinical variables such as age, sex, time from transplant to weaning, and tacrolimus concentration, the LASSO method finally selected 11 predictors...".

We thank the reviewer for prompting us to highlight this point.

A total of 11 predictors were selected by LASSO, how reliable would these selected predictors be with the very limited sample size of 45 (17 TOL vs. 28 non-TOL)? The model would be overfitted by including 11 predictors in a model with n=45. To avoid overfitting and remove less impactful predictors, you may use a larger lambda as minimum lambda + 1SE. The final model would have less concern of overfitting with 3-4 selected predictors based on n=45.

We thank the reviewer for the valuable suggestion to consider the 1-SE (1SE) rule to select a more parsimonious model with fewer predictors while maintaining cross-validated error within one standard error of the minimum rule (λmin), to reduce overfitting. In response, we fit a LASSO logistic regression model using 10-fold cross-validation and compared the models obtained via λmin and 1SE.

The model selected using λmin (λ = 0.86) included 11 predictors, with strong performance (Balanced Accuracy: 0.858, AUC: 0.87 [95% CI: 0.76–0.98], Kappa: 0.716). As suggested, we also evaluated the model at 1SE (λ = 0.98), which included only 4 predictors. This simpler model showed a notable reduction in classification performance (Balanced Accuracy: 0.686, AUC: 0.725 [95% CI: 0.675–0.77], Kappa: 0.337), despite retaining the biomarker of primary interest, and maintains cross-validated error within one standard error of the minimum.

While the 1SE model offers increased simplicity and lower overfitting risk, the λmin model demonstrated superior predictive accuracy and discriminative ability. We therefore report both models, emphasizing that the biomarker of interest remains selected in the reduced model, but highlighting the trade-off between model sparsity and predictive performance.

We have added these comparisons, metrics, and model details to the revised main manuscript [in Results, Methods and Discussion], and included a figure showing cross-validated error across the lambda path (Supplementary Methods and Supplementary Figure 4).

As suggested, we evaluated a more parsimonious model using the 1-SE rule, which selected λ = 0.98. This model includes 4 predictors — including our primary biomarker of interest — and maintains cross-validated error within one standard error of the minimum (λ = 0.86). Although the simplified model shows slightly lower classification performance (e.g., AUC = 0.725 vs. 0.87), it offers a favorable trade-off in terms of model simplicity and potential for generalization, in line with the reviewer’s concerns about overfitting.

To address the concern regarding overfitting, we evaluated model stability and generalizability using multiple complementary approaches:

(1) Cross-Validation Error Variance: We conducted 10-fold cross-validation, and the cross-validated error at the optimal lambda (λ = 0.86) was 1.093 ± 0.02. The low standard deviation (~1.8% of the mean) suggests the model performance is consistent across folds, indicating low variance and a stable model.

(2) Performance Gap between Training and Validation AUC: We also estimated learning curves to assess generalization. The AUC on the training data was 0.84, while the AUC on the held-out validation data was 0.76, yielding a gap of 0.08. This gap is modest and within acceptable limits in biomedical classification contexts, suggesting that overfitting is minimal and that the model generalizes reasonably well to unseen data.

The predictive capacity and performance of the model with selected predictors should be examined and reported

We thank the Reviewer for this important suggestion. In response, we have now explicitly reported the predictive capacity of the LASSO logistic regression models using multiple relevant metrics: the Area Under the Curve (AUC) with 95% confidence intervals, as well as sensitivity, specificity, positive predictive value, and negative predictive value.

To assess overall model performance, we additionally report the overall error rate, balanced accuracy, and Cohen’s kappa coefficient. These metrics are now summarized in the revised Table 2, allowing for a comprehensive evaluation of both the full model (λmin) and the reduced, more parsimonious model (λ1SE).

Yours faithfully,

Dr. José Antonio Pons, on behalf of all authors.

---

## [Decision Letter · Decision Letter 2]

3 Oct 2025

Dear Dr. Pons,

Thank you for submitting your manuscript to PLOS ONE. After careful consideration, we feel that it has merit but does not fully meet PLOS ONE’s publication criteria as it currently stands. Therefore, we invite you to submit a revised version of the manuscript that addresses the points raised during the review process.

The manuscript has been evaluated by two reviewers, and their comments are available below. Reviewer 3 has raised a final concern that need attention. Could you please revise the manuscript to carefully address the concern raised?

We look forward to receiving your revised manuscript.

Kind regards,

Johanna Pruller, Ph.D.

Senior Editor

PLOS ONE

Journal Requirements:

Reviewers' comments:

Reviewer's Responses to Questions

**Comments to the Author**

Reviewer #2: All comments have been addressed

Reviewer #3: (No Response)

2. Is the manuscript technically sound, and do the data support the conclusions?

Reviewer #2: Yes

Reviewer #3: (No Response)

3. Has the statistical analysis been performed appropriately and rigorously?

Reviewer #2: Yes

Reviewer #3: (No Response)

4. Have the authors made all data underlying the findings in their manuscript fully available?

Reviewer #2: Yes

Reviewer #3: (No Response)

5. Is the manuscript presented in an intelligible fashion and written in standard English?

Reviewer #2: Yes

Reviewer #3: (No Response)

Reviewer #2: The latest manuscript provides a full comprehensive evaluation of the stadistic model used to detect the variables that predict OT accurately and adds an alternative simplified model for further generalization.

Reviewer #3: Table 2, the numbers in the classification performance metrics should be shown in 0-100% or 0-1 decimal.

**Do you want your identity to be public for this peer review?** For information about this choice, including consent withdrawal, please see our Privacy Policy

Reviewer #2: No

Reviewer #3: No

---

## [Author Response · Author response to Decision Letter 3]

6 Oct 2025

Manuscript Reference PONE-D-25-05354R2

Point-by-point Rebuttal Letter

Dear Editor and Reviewers,

We sincerely thank Reviewer 2 for their constructive feedback and for accepting our responses and revisions to the previous comments raised. We are now addressing the last concern expressed by the Reviewer 3, as indicated by the Editor. Below, we provide a point-by-point response to the reviewer’s remark, incorporating clarifications and modifications to the manuscript where applicable. Changes in the revised manuscript are displayed as "tracked changes". Also, a clean version of the manuscript has been submitted.

Reviewer #3:

Table 2, the numbers in the classification performance metrics should be shown in 0-100% or 0-1 decimal.

We thank the reviewer for this valuable observation. The values in Table 2 have now been corrected and are presented in decimal format (0–1), as suggested.

Yours faithfully,

Dr. José Antonio Pons, on behalf of all authors.

---

## [Decision Letter · Decision Letter 3]

27 Nov 2025

Longitudinal biomarker progression and validation for predicting operational tolerance in a prospective multicenter liver transplantation immunosuppression withdrawal trial

PONE-D-25-05354R3

Dear Dr. Pons,

We’re pleased to inform you that this revised version of your manuscript (Revision 3) has been judged scientifically suitable for publication and will be formally accepted for publication once it meets all outstanding technical requirements.

With best regards,

Lucienne Chatenoud

Academic Editor

PLOS ONE

Additional Editor Comments (optional):

Reviewers' comments:

Reviewer's Responses to Questions

**Comments to the Author**

Reviewer #2: All comments have been addressed

Reviewer #3: All comments have been addressed

2. Is the manuscript technically sound, and do the data support the conclusions?

Reviewer #2: Yes

Reviewer #3: (No Response)

3. Has the statistical analysis been performed appropriately and rigorously?

Reviewer #2: Yes

Reviewer #3: (No Response)

4. Have the authors made all data underlying the findings in their manuscript fully available?

Reviewer #2: Yes

Reviewer #3: (No Response)

5. Is the manuscript presented in an intelligible fashion and written in standard English?

Reviewer #2: Yes

Reviewer #3: (No Response)

Reviewer #2: All the comments have been addressed. The last version is clear. Congratulations to the authors for the research.

Reviewer #3: (No Response)

**Do you want your identity to be public for this peer review?** For information about this choice, including consent withdrawal, please see our Privacy Policy

Reviewer #2: No

Reviewer #3: No

---

## [Editor Report · Acceptance letter]

PONE-D-25-05354R3

PLOS ONE

Dear Dr. Pons,

I'm pleased to inform you that your manuscript has been deemed suitable for publication in PLOS ONE. Congratulations! Your manuscript is now being handed over to our production team.

Kind regards,

on behalf of

Professor Lucienne Chatenoud

Academic Editor

PLOS ONE